# Sentinel Lymph Node Biopsy in Atypical Spitz Tumor: A Systematic Review

**DOI:** 10.3390/jcm13113232

**Published:** 2024-05-30

**Authors:** Marcodomenico Mazza, Francesco Cavallin, Elisa Galasso, Paolo Del Fiore, Rocco Cappellesso, Fortunato Cassalia, Saveria Tropea, Irene Russo, Mauro Alaibac, Simone Mocellin

**Affiliations:** 1Soft-Tissue, Peritoneum and Melanoma Surgical Oncology Unit, Veneto Institute of Oncology IOV-IRCCS, 35128 Padova, Italy; marcodomenico.mazza@iov.veneto.it (M.M.); saveria.tropea@iov.veneto.it (S.T.); irene.russo@iov.veneto.it (I.R.); simone.mocellin@unipd.it (S.M.); 2Independent Statistician, 36020 Solagna, Italy; cescocava@libero.it; 3Department of Surgical, Oncological and Gastroenterological Sciences (DISCOG), University of Padua, 35128 Padova, Italymauro.alaibac@unipd.it (M.A.); 4Pathological Anatomy Unit, Padua University Hospital, 35121 Padua, Italy; 5Unit of Dermatology, Department of Medicine, University of Padova, 35128 Padua, Italy; fortunato.cassalia@studenti.unipd.it

**Keywords:** skin neoplasm, spitz tumor, atypical, sentinel lymph node biopsy, systematic review

## Abstract

Background: Atypical Spitz tumor (AST) is an intermediate category among Spitz melanocytic neoplasms. Sentinel node biopsy (SNB) has been proposed in the clinical management of AST patients, but this approach remains the subject of debate. This systematic review aims to summarize the available evidence on SNB procedures in AST patients. Methods: A comprehensive search was conducted, including MEDLINE/Pubmed, EMBASE, and SCOPUS, through April 2023. Case series, cohort studies, and case–control studies of AST patients were eligible for inclusion. PRISMA guidelines were followed. Results: Twenty-two studies with a total of 756 AST patients were included. The pooled SNB prevalence was 54% (95% CI 32 to 75%), with substantial heterogeneity (I2 90%). The pooled SNB+ prevalence was 35% (95% CI 25 to 46%) with moderate heterogeneity (I2 39%). Lymphadenectomy was performed in 0–100% of SNB+ patients. Overall survival rates ranged from 93% to 100%, and disease-free survival ranged from 87% to 100% in AST patients. Overall and disease-free survival rates were 100% in SNB patients. Pooled survival estimates were not calculated due to the heterogeneous timing of the survival assessment and/or the small size of the subgroups. All studies clearly reported inclusion criteria and measured the condition in a standard way for all participants, but only 50% indicated valid methods for the identification of the condition. Conclusions: The oncologic behavior of AST is related to an almost always favorable outcome. SNB does not seem to be relevant as a staging or prognostic procedure, and its indication remains debatable and controversial.

## 1. Introduction

Atypical Spitz tumor (AST) is a melanocytic neoplasm of uncertain malignant potential interposed between benign Spitz nevus (SN) and malignant Spitz Melanoma (SM) [1]. The incidence of AST is still uncertain, but it has been estimated to be about 6–8% of the overall number of SN. AST develops in relatively young individuals and is predominant at extremity sites [2]. AST are polymorphic melanocytic tumors typically appearing as nodular lesions and dermoscopically typified by a multicomponent or unspecific pattern. Amelanotic and hypomelanotic nodules with a typical spitzoid pattern under dermoscopy cannot be safely classified as [3]. Histologically, AST combines the distinctive epidermal changes and architectural and cytological features of SN with some worrisome features usually found in melanoma [4]. Differentiating AST from SM with histology can at times be very difficult, if not impossible. Indeed, it is well known that even expert dermatopathologists can fail to reach consensus on diagnosis, discrimination from melanoma, and prediction of outcome using histology alone [5,6]. Only recently, advances in knowledge of the molecular landscape of the melanocytic Spitz lineage have enabled the number of uncertain diagnoses to be reduced [7]. As compared to SN, malignant lesions could genetically present a different set of driver alterations that include *HRAS* or *MAP2K1* mutations, BRAF fusion kinases, and fusion kinases involving *Alk*, *Ros1*, *Ntrk1*, *Ntrk3*, *Met*, *Ret*, *Braf*, or *Map3k8*. Additional alterations involved in the malignant evolution are more than one chromosomal abnormality, loss of 9p21, gains of 6p25, *Tert* promoter, and *TP53* mutations. By immunohistochemistry, SM may exhibit Hmb-45 irregular staining, a high Ki-67 proliferation rate, loss of staining for p16, and PRAME positivity [8,9,10,11]. Comparative genomic hybridization, fluorescence in situ hybridization (FISH), and gene expression profiling can be used to help diagnose SM, but prognostic data for AST are still limited [12].

The clinical management of AST remains extremely challenging, mainly because of the lack of standardized classification criteria and procedures [13]. Sentinel node biopsy (SNB) is the standard of care in staging melanoma patients and is the most important factor in prognosis [14]. SNB has also been proposed in the clinical management of AST, mostly to help the differential diagnosis of ambiguous cases: if tumor cells are found in the SNB, the AST is considered to be melanoma [15,16]. However, this approach to AST patients remains the subject of debate. Some studies have demonstrated that the presence of melanocytes in a sentinel node does not constitute sufficient proof of the malignant nature of the primary tumor [17]. Indeed, despite lymph node involvement, the majority of AST patients are reported to survive without widespread distant metastases, although rare fatal cases have been documented [2,18]. Overall, the mortality rate for AST patients is presumed to be less than 5% [19].

This systematic review aims to summarize the available evidence on SNB procedures in AST patients in order to help clinicians in the diagnostic process and improve patient care.

## 2. Materials and Methods

Study design

This is a systematic review of studies reporting SNB procedures in AST patients. The review was conducted according to the Preferred Reporting Items for Systematic Reviews and Meta-Analyses (PRISMA) guidelines [20]. The review protocol was registered in PROSPERO (CRD42023427924).

Search strategy

MEDLINE/PubMed, EMBASE, and SCOPUS were systematically searched to identify eligible studies. The search was performed without language restrictions through April 2023. In PubMed, the following search strategy was used: atypical spitz AND sentinel lymph node biopsy. The search strategy was tailored to fit other electronic sources. The lists from each source were combined, and the duplicates were removed. Two investigators (MM, EG) separately evaluated the titles and abstracts of the articles and removed those outside the scope of the review. The full-text versions of all potentially eligible articles were examined to remove those not fulfilling the inclusion criteria. Finally, the reference lists of the included articles were searched manually to identify any further studies of interest. Any disagreement was solved by consensus with a third investigator (PDF). Case series, cohort studies, and case–control studies reporting information on SNB in AST patients were eligible for inclusion. Studies not including human subjects were excluded.

Data collection

Two investigators (MM, EG) independently extracted relevant data from the included articles. For each article, the study features (year of publication, study design, sample size, study period), patient and tumor characteristics (sex, age, tumor location, tumor diameter, Breslow thickness, number of mitoses), treatment information (SNB and lymphadenectomy) and outcome measures (overall survival OS, disease-free survival DFS) were collected. A third investigator (PDF) checked the extracted data, and any inconsistency was solved by consensus.

Assessment of the quality of the included studies

Two investigators (MM, EG) independently assessed the quality of the included studies according to the critical appraisal tool of the Joanna Briggs Institute (JBI) [21]. A third investigator (FC) checked the assessments, and any inconsistency was solved by consensus. The JBI tool includes the following 10 items: clear criteria for inclusion; condition measured in a standard, reliable way for all participants; valid methods for identification of the condition for all participants; consecutive inclusion of participants; complete inclusion of participants; clear reporting of the demographics of the participants; clear reporting of clinical information of the participants; outcomes or follow-up results clearly reported; clear reporting of the presenting sites’/clinics’ demographic information; appropriate statistical analysis. Of note, the last item was not relevant for the purpose of our systematic review.

Data synthesis

Meta-analyses of proportions were performed using generalized linear mixed models, and pooled estimates were calculated using the random-effect approach because of the expected heterogeneity across studies. Effect sizes were reported as a proportion with a 95% confidence interval (CI). Heterogeneity was quantified through I2 statistics; of note, high I2 does not imply data inconsistency in a proportional meta-analysis because heterogeneity is expected when assessing prevalence/incidence due to differences in study settings [22]. Publication bias was not assessed with analytical procedures because the underlined assumption (positive findings are published more often) may not hold for proportional studies, where there is uncertainty about the definition of a positive result [22]. Post-hoc sensitivity analyses, including (i) the studies clearly indicating a valid method for the identification of the condition and (ii) the studies clearly indicating the inclusion of consecutive patients, were also performed. Statistical analysis was carried out using R 4.3 (R Foundation for Statistical Computing, Vienna, Austria) [23].

## 3. Results

### 3.1. Search Results

The search identified 199 non-duplicated articles. After excluding 142 articles based on title/abstract, 57 potentially eligible articles were retrieved for full-text review. Of these, 37 were excluded due to a different design (n = 20), a different topic (n = 11), or different participants (n = 4). Another was excluded because the full-text version could not be found. Three additional articles were identified via manual search. Finally, 22 articles [9,10,15,19,24,25,26,27,28,29,30,31,32,33,34,35,36,37,38,39,40,41] were included in the synthesis (Figure 1).

### 3.2. Study Characteristics

The patient and tumor characteristics of the included studies are summarized in Table 1. All studies were case series, including a total of 756 patients (min 9–max 144) aged 1–77 years. Twenty studies specified the tumor location, with the lower limb as the most frequently reported location (12/20, 60%). Tumor diameter was reported in six studies (ranging from 2.3 to 30 mm), Breslow thickness in 18 studies (ranging from 0.3 to 30 mm), and number of mitoses in 10 studies (ranging from 0 to 12 per mm^2^). Appendix A reports how ATS patients were defined in the included studies, and Appendix A summarizes the molecular analyses that were carried out in the included studies.

### 3.3. SNB among AST Patients

Fourteen studies [9,10,15,19,24,25,26,27,29,32,33,34,35,36] included both patients undergoing SNB and those not undergoing SNB, allowing us to calculate the proportion of patients undergoing SNB among AST patients. This proportion ranged from 4% to 100% among the studies, and the pooled proportion was 54% (95% CI 32 to 75%) with substantial heterogeneity (I2 90%) (Figure 2).

### 3.4. Positive SNB among SNB Patients

All but one study [9,10,15,19,24,25,26,27,28,29,30,31,32,33,34,35,36,38,39,40,41] included both patients with positive SNB and those with negative SNB, allowing us to calculate the proportion of patients with positive SNB among those undergoing SNB. This proportion ranged from 0% to 100% among the studies, and the pooled proportion was 35% (95% CI 25 to 46%) with moderate heterogeneity (I2 35%) (Figure 3).

### 3.5. Lymphadenectomy among Positive SNB Patients

In the 19 studies [15,19,24,25,26,27,28,29,30,32,33,34,35,36,37,38,39,40,41] reporting data on lymphadenectomy among positive SNB patients, the proportion of patients undergoing lymphadenectomy among those with positive SNB ranged from 0% to 100% among the studies. In the 17 studies [15,19,24,25,26,28,29,30,32,33,35,36,37,38,39,40,41] including patients undergoing lymphadenectomy, the proportion of patients with positive lymphadenectomy ranged from 0% to 33% among the studies. Pooled estimates were not calculated due to the small size of the subgroups.

### 3.6. Survival

Follow-up information is shown in Table 1. In the 21 studies [9,10,15,19,24,25,26,27,28,29,30,31,32,33,34,36,37,38,39,40,41] reporting survival data, OS ranged from 93% to 100% and DFS from 87% to 100% in AST patients. In the 17 studies [10,15,19,24,26,27,28,29,30,31,32,36,37,38,39,40,41] reporting survival data for SNB patients, OS and DFS were 100% in SNB patients. In the three studies [24,29,36] reporting survival data for at least 10 patients undergoing SNB and 10 patients not undergoing SNB, OS and DFS were 100% in patients undergoing SNB and 87–100% in those not undergoing SNB. Pooled estimates were not calculated due to the heterogeneous timing of survival assessment and/or the small size of the subgroups.

### 3.7. Critical Appraisal of the Quality of Included Studies

Table 2 summarizes the quality assessment of the included studies. All the studies clearly reported the criteria for inclusion of the patients (diagnosis of AST) and measured the condition in a standard way for all participants. However, half of the studies did not clearly indicate a valid method for the identification of the condition (each description is reported in Appendix A). All the studies included every participant who was selected according to the inclusion criteria, but the consecutive inclusion of patients was not clearly stated in 12/22 studies (55%). Most studies clearly reported demographics (20/22, 91%), clinical information (19/22, 86%), and follow-up data (19/22, 86%) and described the study sample with sufficient details to allow comparison with the population of interest (17/22, 77%).

### 3.8. Post-Hoc Sensitivity Analysis

In the studies clearly indicating a valid method for the identification of the condition, the pooled proportion of AST patients undergoing SNB was 63% (95% CI 40 to 81%) (with substantial heterogeneity I2 84%) and the pooled proportion of SNB+ patients among those undergoing SNB was 36% (95% CI 19 to 57%) (with moderate heterogeneity I2 32%).

In the studies clearly indicating the inclusion of consecutive patients, the pooled proportion of AST patients undergoing SNB was 53% (95% CI 31 to 74%) (with substantial heterogeneity I2 77%), and the pooled proportion of SNB+ patients among those undergoing SNB was 25% (95% CI 12 to 46%) (with moderate heterogeneity I2 56%).

## 4. Discussion

In this systematic review, we summarize the available evidence on SNB procedures in AST patients to help clinicians in the management of this neoplasm. SNB is performed in the management of different tumors (such as melanoma or breast cancer) in order to obtain valid information for staging and prognosis. In recent decades, SNB has also been proposed for AST patients, but the borderline behaviors of these lesions may not justify the increased surgical risk of the procedure. Although SNB is considered a minimally invasive procedure, complication rates can be as high as 30% [42]. A biopsy of axillary, inguinal, or laterocervical nodes exposes the patient to the risks of deeper anesthesia and complications such as seromas, surgical site infections, or hematomas [43]. Previously, Lallas et al. [18] performed a literature review on the role of SNB in AST patients and could not find any prognostic benefits associated with this procedure. However, the features of the included studies prevented Lallas et al. from performing a quantitative synthesis of the data. Therefore, the authors concluded that the topic could be considered unresolved at that time. Our review added further information from relevant studies published in the last ten years and refined the inclusion criteria to focus on case series reporting SNB procedures in AST patients. Considering the disagreement existing among clinicians and histopathologists regarding the diagnosis and treatment of AST [44], we believe that an improved definition of therapeutic management is needed.

The difficulties in the correct classification of Spitz tumors are due to the heterogeneous sample size and the variable definition of AST in the included studies. In fact, researchers have published small- to large-sized studies [19,33], which only hint at the magnitude of AST. Furthermore, the definition of AST patients by the clinicians was unclear in many studies due to the extraction of cases from a local database without a correct explanation of the criteria used in the histopathological diagnosis [15,19,24,27,29,30,36,37,40].

AST is now regarded as a melanocytic tumor characterized by epithelioid or spindle melanocytes with large ground-glass cytoplasm along with some alarming features such as increased size, asymmetry, ulceration, necrosis, at least partial lack of maturation, mass-forming growth, pagetoid spread, deep extension, cytological pleomorphism, and an atypical, deep-sited, and increased number of mitoses [1]. This large number of variables accounts for the subjectivity of AST diagnosis and the high rate of inter-observer discrepancy.

In our review, the histopathologic criteria remain poorly predictive in differentiating benign from malignant behavior. Recent studies have suggested various genomic markers for improving the classification, diagnostic agreement, and prognosis of Spitz lesions [6,7]. However, this was outside the scope of this review, which instead focused on SNB.

Since first described by Sophie Spitz in 1948 [45], a wide excision with clear surgical margins is usually performed by most centers [45], and all included studies shared this indication, but the extension of the excision (5–10 mm) is still debated. The role of SNB in AST is controversial, as confirmed by the literature that indicated that it was performed in half of the patients, with substantial heterogeneity among studies showing local preferences and indications. In addition, SNB revealed lymph node involvement in one out of three cases with moderate heterogeneity among studies, suggesting that this procedure may not be necessary in such patients. Identifying AST patients who can benefit from SNB may provide some useful indications, but only two of the included studies investigated the risk factors for positive SNB [36,39]. Urso et al. [39] did not find any risk factors, while Ludgate et al. found that positive SNB patients were younger [36]. Similarly, a greater proportion of nodal micrometastases is reported in younger melanoma patients, and the recommendations include a more competent immune system capable of eliminating deposits and preventing further spreading [46]. Indeed, despite having a greater proportion of positive SNB, younger melanoma patients usually have a better prognosis [47]. Finally, extreme variability was observed regarding the inclusion of lymphadenectomy in the treatment of AST, but the small number of treated patients prevents us from making any reasonable assumptions on this aspect. Generally, AST patients had good outcomes in terms of overall and disease-free survival [9,10,15,19,24,25,26,27,28,29,30,31,32,33,34,36,37,38,39,40,41]. Unfortunately, only a few studies [24,29,36] included sufficient data for investigating the prognostic role of SNB, which remains unclear. Nonetheless, information from such studies may suggest minimal to no survival benefit of SNB in AST patients, but further studies are required to assess this aspect.

This systematic review has some limitations that should be considered by the reader. First, the inclusion of small-sized studies implies the need for caution in the interpretation of the findings. Second, half of the studies did not clearly indicate a valid method for the identification of ATS patients. In addition, half of the studies did not clearly state the inclusion of consecutive patients. Third, pooled estimates regarding lymphadenectomy and survival could not be calculated due to the small size of the subgroups under evaluation. Lastly, the vast majority of the studies do not incorporate the recent molecular data regarding the Spitz lineage of differentiation; thus, this aspect cannot be evaluated and might have affected the results.

## 5. Conclusions

The evidence from the literature suggests the need for a stricter definition of AST, as well as using molecular analyses and better indications on how it should be managed. Nevertheless, SNB procedures in the therapeutic process of AST patients are heterogeneous and do not seem to provide any advantages in tumor staging, since the oncologic behavior of AST is related to an almost always favorable outcome. Histopathologically, differential diagnosis between AST and spitzoid melanomas could play a key role in the allocation of resources and the prognosis. In addition, molecular analysis can be decisive in correctly defining tumor characteristics. We believe that molecular analysis of at least BRAF and NRAS should be recommended to reduce misdiagnosis and/or overtreatment. The indication of SNB in patients with AST remains debated and controversial, and further large studies with homogeneous definitions are needed to define practical guidelines for the clinical management of AST.

## Figures and Tables

**Figure 1 jcm-13-03232-f001:**
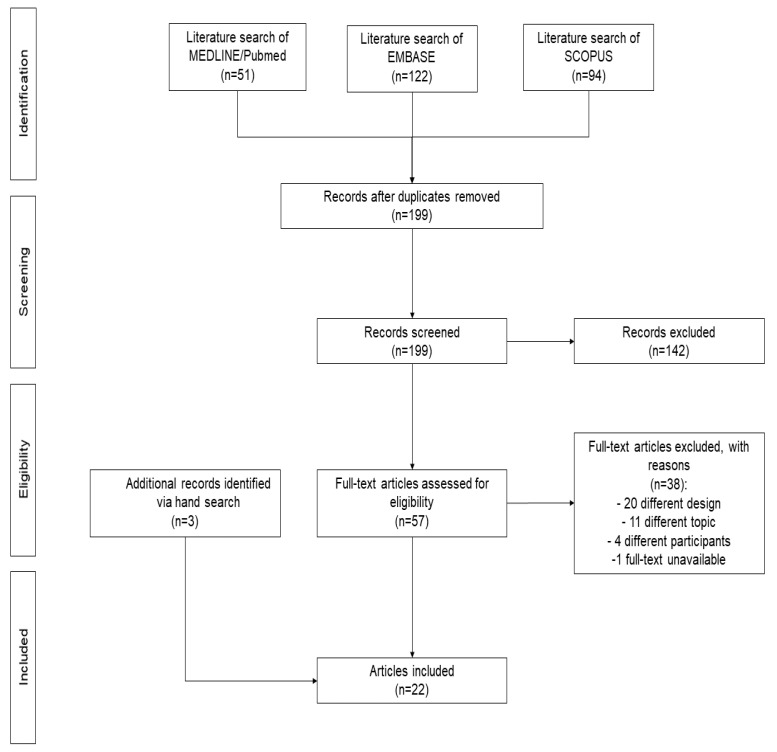
Flow-chart of selection process.

**Figure 2 jcm-13-03232-f002:**
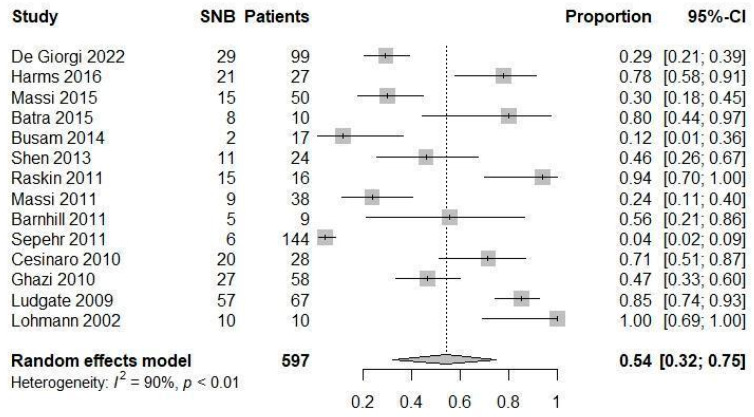
Forest plot: prevalence of patients undergoing SNB among ATS patients [9,10,15,19,24,25,26,27,29,32,33,34,35,36].

**Figure 3 jcm-13-03232-f003:**
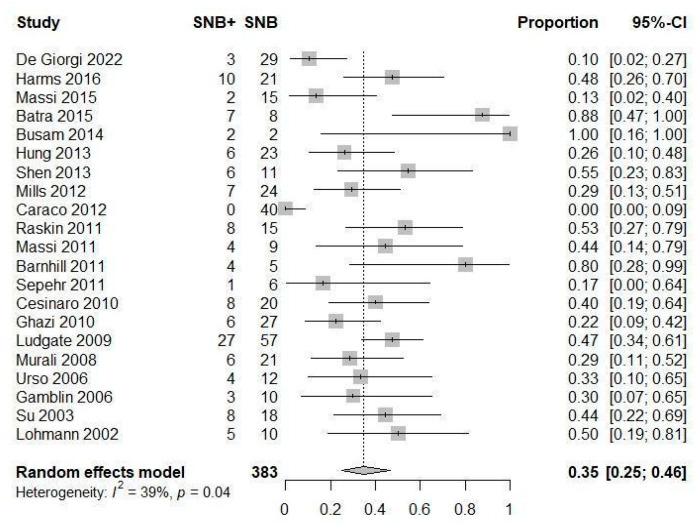
Forest plot: prevalence of SNB+ patients among those undergoing SNB [9,10,15,19,24,25,26,27,28,29,30,31,32,33,34,35,36,38,39,40,41].

**Table 1 jcm-13-03232-t001:** Patient and tumor characteristics of included studies.

First Author	Year	Study Design	N pts	Age at Diagnosis, Years: Mean or Median (Range)	Males (%)	Period of Diagnosis	Prevalent Tumor Site	Diameter, mm: Mean (Range)	Breslow, mm: Mean (Range)	Mitoses per mm^2^: Mean (Range)	Follow-Up, Months: Median (Range)
De Giorgi V [24]	2022	Case series	99	28.1 (2–70)	67	2004–2021	Legs	NA	NA	NA	78 (6–216)
Harms PW [10]	2016	Case series	27	20 (1–65)	NA	NA	NA	NA	2.9 (0.8–7)	NA	31 (4–159)
Massi D [25]	2015	Case series	50	8.5 (1–18)	62	NA	Lower limb	7.2 (2.3–30)	7.2 (2.3–30)	NA	37 (1–300)
Batra S [26]	2015	Case series	10	12.5 (2–16)	60	1980–2014	Head/neck	NA	NA	3 (3–12)	34 (12–84)
Busam KJ [27]	2014	Case series	17	16 (2–35)	53	NA	Extremities	NA	2.9 (1.1–6)	NA	NA (2–48)
Hung T [28]	2013	Case series	23	27 (5–60)	30	1998–2008	Extremities	NA	1.7 (0.8–2.6)	2 (2–8)	57 (2–144)
Shen L [29]	2013	Case series	24	16 (2–56)	30	NA	Limbs	NA	2.2 (0.35–3.8)	2 (0–6)	22 (2–90)
Mills OL [30]	2012	Case series	24	15.5 (4–21)	54	1992–2009	Lower limb; trunk	NA	2.8 (0.5–5.6)	1 (0–6)	49 (2–102)
Caracò C [31]	2012	Case series	40	33 (11–65)	40	2003–2011	Extremities	<10	1.5 (0.8–11)	NA ^a^	46 (16–103)
Raskin L [32]	2011	Case series	16	17.5 (5–65)	37	1999–2009	Lower limb	NA	3 (1.1–6.5)	NA	NA
Massi D [9]	2011	Case series	38	24 (1–53)	45	NA	Lower limb	7.3 (2.5–17)	2.1 (1–15)	1.0 (0–12)	71 (8–156)
Barnhill RL [33]	2011	Case series	9	18 (6–40)	44	2006–2010	Neck	6.5 (3.5–10)	3.6 (0.66–5.35)	2.4 (1–5)	22 (6–50)
Sepehr A [19]	2011	Case series	144	30.2 (3–77)	40	1987–2002	Lower limb	NA	NA	NA	109 (1–206)
Cesinaro AM [34]	2010	Case series	28	32 (3–56)	35	NA	Lower limb	8.6 (5–17)	4.1 (1–12)	NA	54 (4–156)
Ghazi B [35]	2010	Case series	58	24 (6–60)	41	1992–2007	NA	NA	NA	NA	56 (1–160)
Ludgate MW [36]	2009	Case series	67	23.7 (1.7–65)	39	1994–2007	Lower limb	NA	2.4 (0.3–8)	NA ^b^	43 (32–57)
Busam KJ [37]	2009	Case series	11	11.5 (6–17)	46	1949–2005	Lower limb	NA	4.6 (2.1–12)	3 (1–10)	61 (36–132)
Murali R [38]	2008	Case series	21	25 (6–50)	43	1999–2006	Lower limb	NA	2.4 (0.5–5.5)	3.2 (0–10)	21 (1–61)
Urso C [39]	2006	Case series	12	23 (2–48)	25	1998–2005	Lower limb	7 (5–9)	3.4 (1.12–5.7)	NA	26 (2–90)
Gamblin TC [40]	2006	Case series	10	22 (7–47)	50	NA	Lower limb	NA	4.1 (1–12)	NA	28 (13–57)
Su LD [41]	2003	case series	18	16 (5–32)	33	1998–2001	Lower limb	NA	3.5 (1.2–7.9)	1 (1–7)	12 (3–42)
Lohmann CM [15]	2002	Case series	10	21 (7–46)	30	NA	Head and neck	NA	4.8 (0.35–7)	2.8 (0–6)	34 (10–54)

NA: not available. ^a^ Mitoses present in 16 patients, but numerical data not reported. ^b^ Mitoses present in 47 patients, but numerical data not reported.

**Table 2 jcm-13-03232-t002:** Summary of the quality assessment of the included studies (assessed using the critical appraisal tool of the Joanna Briggs Institute).

First Author	Year	Clear Criteria for Inclusion	Condition Measured in a Standard, Reliable Way for All Participants	Valid Methods for Identification of the Condition for All Participants	Consecutive Inclusion of Participant	Complete Inclusion of Participants	Clear Reporting of the Demographics of the Participants	Clear Reporting of Clinical Information of the Participants	Outcomes or Follow-Up Results Clearly Reported	Clear Reporting of the Presenting Sites’/Clinics’ Demographic Information	Appropriate Statistical Analysis
De Giorgi V [24]	2022	yes	yes	unclear	yes	yes	yes	yes	yes	yes	N/A
Harms PW [10]	2016	yes	yes	yes	unclear	yes	no	no	yes	no	N/A
Massi D [25]	2015	yes	yes	yes	yes	yes	yes	yes	yes	yes	N/A
Batra S [26]	2015	yes	yes	yes	yes	yes	yes	yes	yes	yes	N/A
Busam KJ [27]	2014	yes	yes	unclear	unclear	yes	no	yes	no	unclear	N/A
Hung T [28]	2013	yes	yes	yes	yes	yes	yes	yes	yes	no	N/A
Shen L [29]	2013	yes	yes	unclear	yes	yes	yes	yes	yes	no	N/A
Mills OL [30]	2012	yes	yes	unclear	yes	yes	yes	yes	yes	yes	N/A
Caracò C [31]	2012	yes	yes	yes	yes	yes	yes	yes	yes	yes	N/A
Raskin L [32]	2011	yes	yes	yes	yes	yes	yes	yes	yes	yes	N/A
Massi D [9]	2011	yes	yes	yes	unclear	yes	yes	yes	yes	yes	N/A
Barnhill RL [33]	2011	yes	yes	yes	unclear	yes	yes	yes	yes	yes	N/A
Sepehr A [19]	2011	yes	yes	unclear	unclear	yes	yes	yes	yes	no	N/A
Cesinaro AM [34]	2010	yes	yes	yes	unclear	yes	yes	yes	no	yes	N/A
Ghazi B [35]	2010	yes	yes	unclear	yes	yes	yes	yes	yes	yes	N/A
Ludgate MW [36]	2009	yes	yes	unclear	unclear	yes	yes	yes	yes	yes	N/A
Busam KJ [37]	2009	yes	yes	unclear	unclear	yes	yes	yes	yes	yes	N/A
Murali R [38]	2008	yes	yes	unclear	yes	yes	yes	no	no	yes	N/A
Urso C [39]	2006	yes	yes	yes	unclear	yes	yes	yes	yes	yes	N/A
Gamblin TC [40]	2006	yes	yes	unclear	unclear	yes	yes	yes	yes	yes	N/A
Su LD [41]	2003	yes	yes	yes	unclear	yes	yes	no	yes	yes	N/A
Lohmann CM [15]	2002	yes	yes	unclear	unclear	yes	yes	yes	yes	yes	N/A

## Data Availability

All data generated during and/or analyzed during this study are available within the paper.

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
