# Peer review of "Sentinel Lymph Node Biopsy in Atypical Spitz Tumor: A Systematic Review"

_jcm, 2024, doi:10.3390/jcm13113232_

Round 1
Reviewer 1 Report
Comments and Suggestions for Authors
The manuscript is clear and relevant to the field. The cited sources are appropriate because this is a meta-analysis.
The manuscript has a solid scientific basis, but this area should be further analyzed and researched. The test results are adequate for the method. The tables are properly constructed. The conclusions are consistent
- The topic of the study is not new, but it is important for diagnostic purposes. The aim of this review is to analyze the available studies on the SNB procedure in patients with AST.
- The study is rather a review of the available literature on whether node biopsy in patients with diagnosed Spitz is important for treatment or diagnosis. This is not a new topic, but it confirms earlier reports. the research methodology is appropriate because it is a review of reports, so the author used the appropriate methodology in this pre-land.
- Research conclusions show that the SNB procedure in the therapeutic process of patients with AST does not provide adequate benefits in assessing the severity of Spitz.
- Research reports are very extensive. The study involves a meta-analysis of 22 studies that included as many as 756 patients with AST. The tables are prepared correctly and are legible and clear.
Author Response
Reviewer #1
The manuscript is clear and relevant to the field. The cited sources are appropriate because this is a meta-analysis.
The manuscript has a solid scientific basis, but this area should be further analyzed and researched. The test results are adequate for the method. The tables are properly constructed. The conclusions are consistent
- The topic of the study is not new, but it is important for diagnostic purposes. The aim of this review is to analyze the available studies on the SNB procedure in patients with AST.
- The study is rather a review of the available literature on whether node biopsy in patients with diagnosed Spitz is important for treatment or diagnosis. This is not a new topic, but it confirms earlier reports. the research methodology is appropriate because it is a review of reports, so the author used the appropriate methodology in this pre-land.
- Research conclusions show that the SNB procedure in the therapeutic process of patients with AST does not provide adequate benefits in assessing the severity of Spitz.
- Research reports are very extensive. The study involves a meta-analysis of 22 studies that included as many as 756 patients with AST. The tables are prepared correctly and are legible and clear.
RE: Thank you for your comment and for the appreciation of the quality of our work.
Reviewer 2 Report
Comments and Suggestions for Authors
I have read this review with great interest, however there are some remarks which should be included before possible approval of the manuscript:
1. Gene names should be written in Italics.
2. Authors should try to present the table the survival from the reliable studies for cohort of SLN-positive and negative cases and group of patients undergoing and not undergoing SLN biopsy.
Comments on the Quality of English Languagenone
Author Response
Reviewer #2
I have read this review with great interest, however there are some remarks which should be included before possible approval of the manuscript:
- Gene names should be written in Italics.
RE: Done as requested
- Authors should try to present the table the survival from the reliable studies for cohort of SLN-positive and negative cases and group of patients undergoing and not undergoing SLN biopsy.
RE: Unfortunately, the included studies did not specify the survival for SLN-positive and SLN-negative cases. In the Result section, we specified the survival for patients undergoing and not undergoing SLN biopsy:” In three studies (24,29,36) reporting survival data for at least 10 patients undergoing SNB and 10 patients not undergoing SNB, OS and DFS were 100% in patients undergoing SNB, and 87-100% in those not undergoing SNB.” (lines 193-195).
Thanks for the comments that sure will improve our review.
Paolo
Reviewer 3 Report
Comments and Suggestions for Authors
The manuscript reports a systematic review of published papers on sentinel lymph node biopsy in atypical Spitz tumors. It analyzes clinical features, the frequency of the SNB procedure and the positivity rate of SNB as well as survival and DFS.
Comments and suggestions:
1) The authors speak of OS and DFS but do not specify the median time of follow up. Therefore the results are of limited interests.
2) The description of the AST is debatable (Typically, it presents as an amelanotic papule or nodule with symmetrical, well-circumscribed raised borders and a shiny stretched epidermis covering the lesion.)
3) Information in the text does not correspond to the cited references (for example: Overall, the mortality rate in 76 AST patients is presumed to be less than 5% (19).)
4) The molecular data on AST in the published papers could be presented more in detail as molecular diagnosis of AST could lead to a more consistent definition of AST which represents the real medical need.
Comments on the Quality of English LanguageThe English language in the manuscript is of modest quality. The manuscript should be corrected by a native speaker.
Author Response
Reviewer #3
1) The authors speak of OS and DFS but do not specify the median time of follow up. Therefore the results are of limited interests.
RE: We thank the Reviewer for the suggestion. In the original version, we advised the reader about the heterogeneous timing of survival assessment which precluded any pooled estimates of survival data (“Pooled estimates were not calculated due to the heterogeneous timing of survival assessment and/or the small size of the subgroups “, lines 197-198), but we agree about the relevance of providing such information. In the revised version, we added the follow-up duration for each study in Table 1.
2) The description of the AST is debatable (Typically, it presents as an amelanotic papule or nodule with symmetrical, well-circumscribed raised borders and a shiny stretched epidermis covering the lesion.)
RE: we have corrected by making this description:” AST are polymorphic melanocytic tumors typically appearing as nodular lesions and dermoscopically typified by a multicomponent or unspecific pattern. Amelanotic and hypomelanotic nodules with a typical spitzoid pattern under dermoscopy cannot be safely classified as SN” (lines 52-55).
3) Information in the text does not correspond to the cited references (for example: Overall, the mortality rate in 76 AST patients is presumed to be less than 5% (19).)
RE: Done as requested we have checked and corrected the numbering of the references.
4) The molecular data on AST in the published papers could be presented more in detail as molecular diagnosis of AST could lead to a more consistent definition of AST which represents the real medical need.
RE: Done as requested we added the typical number of mutations detected by FISH
5) The English language in the manuscript is of modest quality. The manuscript should be corrected by a native speaker.
RE: Done as requested the text was revised by a professional language editor
Thank you for your comments that have improved the quality of our work.
Paolo
Round 2
Reviewer 3 Report
Comments and Suggestions for Authors
The authors have responded to the previous reviewer's comments.
Additional comments:
1) I would suggest to include the landmark publications of the Boris Bastian group on the genetics of spizoid tumors.
2) 3.5 Lymphadenectomy among.... . What is meant by "patients with positive lymphadenectomy".
3) 3.6 Survival. What is meant by "were 100% in SNB patients"? Patients who underwent SNB procedure or Patients with tumor cells within the sentinel? Please specify.
4) Supplementary Table 1. The definition of AST contains the difference between conventional Spitz nevus but the information is lacking in many cases, on what ground a spizoid melanocytic proliferation is diagnosed in the first place.
Author Response
Reviewer #3
1) I would suggest to include the landmark publications of the Boris Bastian group on the genetics of spizoid tumors.
RE: Thank you for the correct indication, we have cited this publication in the text (ref.11) and bibliography: Roy SF, Milante R, Pissaloux D, Tirode F, Bastian BC, Fouchardière A, Yeh I. Spectrum of Melanocytic Tumors Harboring BRAF Gene Fusions: 58 Cases With Histomorphologic and Genetic Correlations. Mod Pathol. 2023 Jun;36(6):100149. doi: 10.1016/j.modpat.2023.100149. Epub 2023 Feb 24. PMID: 36841436.
2) 3.5 Lymphadenectomy among.... . What is meant by "patients with positive lymphadenectomy".
RE: the proportion of patients with other positive node at the complete lymphadenectomy ranged from 0% to 33% among the studies.
3) 3.6 Survival. What is meant by "were 100% in SNB patients"? Patients who underwent SNB procedure or Patients with tumor cells within the sentinel? Please specify.
RE: were 100% in patients undergoing SNB procedure.
The survival data between positive SNB patients and patients not undergoing SNB is reported only for three studies.
4) Supplementary Table 1. The definition of AST contains the difference between conventional Spitz nevus but the information is lacking in many cases, on what ground a spizoid melanocytic proliferation is diagnosed in the first place.
RE: In The supplementary table 1 is reported how the authors defined AST tumors. in discussion, we specified how the AST definition of clinician was unclear in many studies, due to the extraction of cases from a local database, without a correct explanation of the criteria used in the histopathological diagnosis.
Thank you for your comments that have improved the quality of our work.
Paolo